# Uracil as a Zn-Binding Bioisostere of the Allergic Benzenesulfonamide in the Design of Quinoline–Uracil Hybrids as Anticancer Carbonic Anhydrase Inhibitors

**DOI:** 10.3390/ph15050494

**Published:** 2022-04-19

**Authors:** Samar A. El-Kalyoubi, Ehab S. Taher, Tarek S. Ibrahim, Mohammed Farrag El-Behairy, Amany M. M. Al-Mahmoudy

**Affiliations:** 1Department of Pharmaceutical Organic Chemistry, Faculty of Pharmacy (Girls), Al-Azhar University, Nasr City, Cairo 11651, Egypt; samarel-kalyoubi.52@azhar.edu.eg; 2Department of Pharmaceutical Organic Chemistry, Faculty of Pharmacy, Al-Azhar University, Assiut 71524, Egypt; ehabtaher@azhar.edu.eg; 3Department of Pharmaceutical Chemistry, Faculty of Pharmacy, King Abdulaziz University, Jeddah 21589, Saudi Arabia; 4Department of Pharmaceutical Organic Chemistry, Faculty of Pharmacy, Zagazig University, Zagazig 44519, Egypt; amanyma@zu.edu.eg; 5Department of Organic and Medicinal Chemistry, Faculty of Pharmacy, University of Sadat City, Menoufiya 32897, Egypt; mohammed.farrag@fop.usc.edu.eg

**Keywords:** uracil, quinoline, carbonic anhydrase, anticancer, zinc-binding group

## Abstract

A series of quinoline–uracil hybrids (**10a–l**) has been rationalized and synthesized. The inhibitory activity against hCA isoforms I, II, IX, and XII was explored. Compounds **10a–l** demonstrated powerful inhibitory activity against all tested hCA isoforms. Compound **10h** displayed the best selectivity profile with good activity. Compound 10d displayed the best activity profile with minimal selectivity. Compound **10l** emerged as the best congener considering both activity (IC_50_ = 140 and 190 nM for hCA IX and hCA XII, respectively) and selectivity (S.I. = 13.20 and 9.75 for II/IX, and II/XII, respectively). The most active hybrids were assayed for antiproliferative and pro-apoptotic activities against MCF-7 and A549. In silico studies, molecular docking, physicochemical parameters, and ADMET analysis were performed to explain the acquired CA inhibitory action of all hybrids. A study of the structure–activity relationship revealed that bulky substituents at uracil *N*-1 were unfavored for activity while substituted quinoline and thiouracil were effective for selectivity.

## 1. Introduction

Carbonic anhydrases (CAs, EC 4.2.1.1) are a superfamily of metalloenzymes that are distributed extensively throughout living organisms [1]. CAs are subclassified to eight gene families (α, β, γ, δ, ζ, η, θ, and ι) [2]. As these enzymes are ubiquitously represented in the body, they are major participants in reaction catalysis and physiological processes. In addition to their physiological impacts, the modulation of the natural function or upregulation of these enzymes demonstrate immense benefits for controlling many pathological conditions [3]. The α-CA enzymes found in mammals are divided into four subgroups consisting of several isoforms. The CA I and CA II isoforms belong to the cytosolic subgroup of α-CA, while CA IX and CA XII isoforms are part of the membrane-associated α-CAs subgroup [4]. For instance, sulfonamides were determined as potent inhibitors of α-CAs and can be used to exploit many clinical applications such as diuresis, antiglaucoma, anticonvulsant, antiobesity, analgesic, and anticancer activities [5].

Cancer one of the leading mortality-causing diseases, competing with cardiovascular diseases for first place. It is also one of the overwhelming barriers to high life expectancy owing to its social and economic consequences [6]. In 2020, 19.3 million people were diagnosed with cancer, with 10.0 million deaths reported. This number is expected to reach 28.4 million in 2040 [7]. Many molecular mechanisms and biological targets are connected to the underlying mechanisms of cancer. Of these, human carbonic anhydrases (hCA) IX and XII have been recognized as tumor-associated proteins in hypoxic and other solid tumors where they actively contribute to the permanence and metastasis of the necrotic cell [8]. Notably, a great number of scientists have emphasized that both are biomarkers and therapeutic targets for various cancer types. Accordingly, dual targeting, i.e., inhibition, of the two latter isozymes represent a remarkable challenge for the development of novel anticancer drugs [9,10,11,12].

Following the successful impact of its modulation, the search for further pharmacophores—other than sulfonamides—that can fit/block the active site of CAs has become an interesting topic in medicinal chemistry research [13,14,15]. Afterwards, many chemical moieties have been revealed as isosteres for sulfonamides, such as hydroxybenzene, fullerenes, benzopyrone, thiobenzopyrone, and boronic acids [3]. These new effective chemical pharmacophores have fostered the ability to design promising pharmacologically active candidates that circumvent sulfonamide side effects. In addition, they support further understanding of the underlying mechanisms of CAs and can be utilized for the rational drug design of novel candidates as CAs modulators.

It is important to indicate that CAs are not the sole Zn-containing enzymes that extensively affect cancer physiology [16]. Histone deacetylases (HDACs) are Zn-containing enzymes that play a crucial role in anticancer therapy [17]. Fortunately, the zinc-binding groups (ZBGs) of HDAC inhibitors are different from those reported for CAs, as per the recent reviews [18,19]. The ZBGs of HDACIs are classified into classical and nonclassical groups. The classical ZBGs, such as hydroxamic acid [20] and benzamide [21], are characterized by potent activity, selectivity, toxicity, and in vivo instability [22]. Carboxylic acid [23] and thiol [24] groups were also used as ZBGs in HDACI design. The nonclassical type include imidazole-thione [25], tropolone derivatives [26], 3-hydroxypyridin-2-thione (3-HPT) [27], chelidamic derivatives [28], benzoylhydrazide [29], trifluoromethyloxadiazolyl (TFMO) [30], 2-(oxazol-2-yl)phenol [31], hydroxypyrimidines [32], and β-hydroxymethyl chalcone [33] (Figure 1).

Quinoline is an abundant pharmacophore in medicinal chemistry and exhibits remarkable pharmacological efficacies such as antituberculosis [34], antimalarial [35], antiviral [36,37], anticancer [35,38], antibacterial [37], antifungal [37,39], antileishmanial [40], and anti-inflammatory [41] activity. Moreover, several quinoline-based candidates showed very promising anticancer activities, including neratinib [42], bosutinib [43], foretinib [44], and topotecan [45], which are currently in clinical trials. Additionally, kinase inhibitors [35,46,47], apoptotic agents [35,48], microtubule-targeting agents [35,49], topoisomerase inhibitors [35,50], epigenetic enzyme inhibitors [35,51], transcription factor inhibitors [35,52], and carbonic anhydrase inhibitors [53] are represented (Figure 2).

Similarly, the pyrimidine moiety is a well-documented pharmacophore in medicinal chemistry [53,54,55,56]. Anticancer pyrimidine-containing marketed drugs exhibit their biological activity via assorted mechanisms, such as tyrosine kinase inhibition (imatinib, dasatinib) [57], the inhibition of DNA synthesis (uracil mustard) [58], thymidylate synthase (fluorouracil) [59], nucleoside metabolic inhibition (gemcitabine, trifluridine) [60], antimetabolite (floxuridine) [61], DNA polymerase inhibition (cytarabine) [62], DNA methyltransferase inhibitor (azacitidine) [63], inducing DNA hypomethylation and corresponding alterations in gene expression (decitabine) [64], and transition state analog inhibition of cytidine deaminase (zebularine) [65] (Figure 3).

Primary benzenesulfonamide derivatives are well known for their CA inhibition, which confers several biological activities, especially anticancer activity. In this regard, quinoline-based sulfonamides have been recently reported with impressive CA inhibition, where compound I displayed inhibitory activity against hCA I and II isoforms with Ki ranges of 0.96–9.09 µM and 83.3–3.59 µM for hCA I and II, respectively [53]. Compound II displayed potent inhibitory activity against the well-known cancer-related isozymes hCA IX and XII with Ki values of 0.019 µM and 0.009 µM, respectively [66].

In all the investigated compounds, the quinoline backbone was important for the biological activity; however, the zinc-binding group (primary benzenesulfonamide) is the essential moiety [53,67,68,69,70]. Despite its key impact, the allergic response to sulfonamide drugs is a very unfavorable reaction [71,72]. Thus, the search for an alternative bioisostere is of much interest in the field of medicinal chemistry. Notably, uracil compounds III and IV are non-sulfonamide derivatives that show promising CA inhibitory activity and were confirmed as leads for generating potent CAIs, triggering further isoforms [73]. From the perspective of the structure–activity relationship, the uracil moiety with C=O, N, and NH_2_ represents an active pharmacophore, rich with binding groups that can substitute the sulfonamide interactions in the receptor (Figure 4).

In light of the abovementioned findings, and by applying the tactic of pharmacophoric hybridization, a novel series of quinoline-based uracil synthetic molecules has been rationalized, synthesized, and tested for their CA inhibitory and anticancer activities. The utilized uracils with diverse functionality were similar to the primary benzenesulfonamide moiety regarding the two sites for both H-bond donors and H-bond acceptors. This rationale has promoted the exploration of such hybrids. These hybrid structures have been lined chemically via an imine bond (Schiff’s base), which has been carefully selected as an anchor based on the reported CA inhibitory activity of its derivatives (compound V) [74] (Figure 4). An additional notable feature of our novel hybrids is the synthesis of quinoline-based thiouracil to realize the power of sulfur-based compounds as a zinc-binding group [75]. Docking and in silico studies have been performed to determine the CA inhibitory profile for the targeted quinolones.

## 2. Results and Discussion

### 2.1. Chemistry

The utilized protocol of our designed targets **10a–****l** consists of two key structural features; first, the quinoline-3-carbaldehyde moiety (Figure 1), and second, the 5,6-diaminouracil/thiouracil pharmacophores (Figure 2). The straightforward synthesis leading to 2-quinolone-3-carbaldehyde derivatives (**4a****–****c**), which has benefit of conciseness, is shown in Figure 1. The acetylation of aniline derivatives **1a****–****c** through glacial acetic acid/acetic anhydride at a temperature of 0 °C furnished the corresponding anilides (**2a****–****c**) [76,77]. These compounds underwent the Vilsmeier–Haack reaction to give the 2-chloroquinoline aldehyde derivatives (**3a****–****c**) [78,79]. The oxidation of the chlorine group at position 2 into a ketone has been successfully performed under reflux in acidic conditions (acetic acid) to deliver the desired synthons (**4a****–****c**). This reaction was initiated by oxidation with acetic acid, then completed by the departure of chlorine as a good leaving group to produce the favored product. The reaction appeared to be facilitated through the electron withdrawing effect of nitrogenous heteroatom [78,79].

As shown in Figure 2, the variable 5,6-diaminouracil/thiouracil derivatives **9a–e** were prepared via consecutive cyclization of *N*-alkylurea/thiourea with ethyl cyanoacetate in the presence of sodium ethoxide, which initially gave 6-aminouracil/thiouracils **7a****–****e**. This was readily followed by conversion via a nitrosation process using nitrous acid and then reduction of nitroso uracil/thiouracils **8****a–****e** via the reducing agent ammonium sulfide [80,81].

Condensation of diamino-uracil/thiouracils **9a–e** with the appropriate quinoline aldehydes **4a–c** under refluxing condition in ethanol for 1 h furnished the anticipated quinoline-based uracils/thiouracils **10a–l** (Figure 3). The structures of these novel compounds were also characterized by using thin-layer chromatography and melting point methods. Our novel later hybrids were determined using ^1^H NMR, ^13^C NMR, elemental, and mass spectroscopic techniques. The ^1^H NMR spectra showed characteristic imino protons (N=CH) of Schiff bases appearing in the range from 8.45–9.91 ppm as sharp singlets. The ^13^C NMR spectra further supported the assigned structures. The ^13^C NMR shift of N=CH carbon atoms appeared in the range from 161.32–164.41 ppm as a singlet signal.

### 2.2. Biology

#### 2.2.1. Carbonic Anhydrase Inhibition

Activities of all CA isoenzymes were estimated following the previously described colorimetric method of Verpoorte et al. (1967) [82,83] using BioVision Carbonic Anhydrase (CA) Inhibitor Screening Kit (Catalog # K473-100). As per Table 1, all candidates **10a****–****l** showed variable inhibitory activities against the tested CA isoforms. For hCAI, all the synthesized candidates **10a****–****l** demonstrated inhibitory activities. Compounds **10a**, **10d**, **10f**, **10j**, and **10l** exhibited inhibitory concentrations in the nanomolar range (230–970 nM), whereas the other compounds showed activity in the micromolar range (Table 1 and Figure 5). Compounds **10d**, **10f**, and **10l** displayed higher inhibitory activity (IC_50_ = 230, 330, and 250 nM, respectively) than the reference standard acetazolamide (AAZ, IC_50_ = 760 nM). It is shown that the placement of small lipophilic methyl groups at *N*-1 uracil (where X = O), i.e., compound **10d**, was preferred to X = S and the bulkier ethyl and benzyl groups.

In the case of the physiologically dominant hCA II, candidates **10a**, **10c**, **10d**, and **10f** showed potent inhibitory activity in the nanomolar range (IC_50_ = 520, 660, 260, and 790 nM, respectively, Table 1 and Figure 5). Similarly, compound **10d** was found to be the most potent hCA II inhibitor with a lower inhibitory concentration (IC_50_ = 260 nM) than the reference standard acetazolamide (AAZ, IC_50_ 390 nM). This highlights the effect of bulkiness on the interaction with the hCA II binding site. Fortunately, tumor-linked hCA IX was efficiently inhibited by most of our candidates in the nanomolar range, while three candidates (**10e**, **10g**, and **10k**) showed inhibition in the low micromolar range (IC_50_ = 1.37, 1.08, and 1.28 µM, respectively). Moreover, compounds **10a**, **10d**, and **10f** displayed superior inhibitory activity against the hCA IX isozyme in the nanomolar range, with IC_50_ values of 250, 220, and 270 nM, respectively. Thiouracil hybrid **10l** represented the best hCA IX inhibitor with a half-inhibitory concentration (IC_50_ = 140 nM) lower than the standard acetazolamide (AAZ, IC_50_ = 150 nM). In addition to its potent activity, hybrid 10l showed higher selectivity (13.25-fold) towards the tumor-linked hCA IX enzyme than the physiologically dominant hCA II and 1.75-fold higher selectivity to hCA I (Table 2 and Figure 6). Similarly, compound **10h** displayed very good selectivity to transmembrane tumor-associated isoform hCA IX. A further tumor-related CA isoform (hCAXII) was potentially inhibited by our quinoline–uracil hybrids. All our synthesized targets **10a****–****l** inhibited hCA XII in the nanomolar range. Compounds **10d**, **10f**, and **10l** (where IC_50_ = 190, 170, and 190 nM, respectively) were more potent than acetazolamide (IC_50_ = 230 nM) while compound **10i** (IC_50_ = 230 nM) was equipotent to the standard acetazolamide.

Regarding selectivity profile, compounds **10g**, **10h**, **10i**, **10k**, and **10l** were the most selective for hCA XII over both hCAI and hCAII (Table 2 and Figure 7). Compound **10h** displayed the best selectivity profile (S.I. = 5.39, 7.52, 8.00, and 11.16 for I/IX, II/IX, I/XII, and II/XII, respectively) with good activity (600 and 400 nM for hCA IX and hCA XII, respectively). However, compound **10d** displayed the best activity profile against both hCA IX and hCA XII (IC_50_ = 220 and 190 nM, respectively) with a minimal selectivity profile (S.I. = 1.09, 1.21, 1.24, and 1.37 for I/IX, II/IX, I/XII, and II/XII, respectively). Herein, compound **10l** emerged as the best congener considering both activity and selectivity. Compound **10l** (IC_50_ = 190 nM) showed better activity than **10h** (IC_50_ = 400 nM) and the same activity as **10d** (IC_50_ = 190 nM) against hCA XII. Further, **10l** (IC_50_ = 140 nM) showed better activity than both **10h** and **10d** against hCAIX (IC_50_ = 600 and 220 nM, respectively). In addition, **10l** is the only candidate with superior activity (IC_50_ = 140 and 190 nM for hCA IX and hCA XII, respectively) in comparison to acetazolamide (AAZ, IC_50_ = 150 and 230 nM for hCA IX and hCA XII, respectively). In addition, **10l** displayed better selectivity than acetazolamide (S.I. = 13.20 and 9.75 for II/IX, and II/XII, respectively) toward the tumor-linked isoforms hCAs IX and XII against physiologically dominant hCA II (Table 2).

#### 2.2.2. Antiproliferative Activity

Based on their activities and selectivity against tumor-linked CA isoforms, compounds **10d** and **10l** were selected for further screening against the breast cancer cell line (MCF-7) (HTB-22 from ATCC, Manassas, VA, USA) and lung cancer cell line (A549) (CCL-185 from ATCC, Manassas, VA, USA) under hypoxic conditions to evaluate their in vitro antiproliferative activity using the MTT assay protocol [84] (Table 3). Both compounds showed activity against the tested cell lines. However, compound 10d had more potent activity than 10l against the MCF7 breast cancer cell line with IC50 = 2.87 ± 0.05 compared with compound 10l IC50 value of 4.08 ± 0.08 µM. For the reference standard staurosporine, the IC50 was 6.92 ± 0.18 µM. Despite their superior activity over staurosporine against breast cancer cell line, compounds **10d** and **10l** had lower activity than staurosporine towards the lung cancer cell line (Figure 7).

#### 2.2.3. Assessment of Apoptotic Marker Levels

The two cytoplasmic proteins, B-cell lymphoma protein 2 (Bcl-2) and its associated X protein (Bax), are essential for apoptosis in normal cells. Bax is a promoter and Bcl-2 is an inhibitor of apoptosis [85]. Thus, the effect of **10d** and **10l** on apoptotic markers Bax and Bcl-2 in a breast cancer cell line (MCF-7) and a lung cancer cell line (A549) has been estimated. Treatment of MCF-7 and A549 cell lines with compounds **10d** and **10l** resulted in the upregulation of Bax levels by nearly six-fold relative to the control while the expression of Bcl-2 levels was downregulated in comparison with the control (Table 3).

### 2.3. In Silico Study

#### 2.3.1. Physicochemical and Pharmacokinetic Parameters

The SWISSADME server [86] was utilized to assess the ability of the investigated compounds (**10a–l**) to act as drugs via estimating the physicochemical and pharmacokinetic properties. No compounds were expected to cross the BBB, inhibit cytochrome enzymes, violate the Lipinski’s rule, or give PAINS alerts. In addition, all compounds (except **10g**, **10h**, and **10l**) demonstrated high GIT absorption. The investigated compounds showed good synthetic accessibility and bioavailability scores (Table 4). Moreover, no compounds inhibited cytochrome P450 enzymes, indicating there was no expected pharmacokinetic-related drug–drug interactions.

#### 2.3.2. Molecular Docking Study

Molecular docking studies were performed to investigate the interactions of the target compounds **10a**–**10l** with the binding site of the human carbonic anhydrases IX using Discovery Studio. hCA IX is considered promising targets for cancer treatment and their inhibition can reduce the growth of primary tumors and metastases. For the CA IX isoform, the PDB file 5FL4 containing hCA IX co-crystallized with 5-(1-naphthalen-1-yl-1,2,3-triazol-4-yl)thiophene-2 sulfonamide was obtained from the Protein Data Bank [87].

The protein structure was prepared by 3D protonation and the water molecule and ligand that were not implicated in the active site were removed. The active site then generated with the default protocol [88]. The 2D and 3D interaction diagrams for the ligand and compounds **10d**, **10f**, and **10l** are shown in Figure 8. Analyzing the ligand–protein interactions can help better understand the selectivity of the compound for hCA IX with respect to the other hCA isoforms.

The co-crystallized ligand forms two hydrogen bonds with Thr 200 and Leu199; it was found to chelate the zinc ion through the sulfonamide group. It formed pi–sulfur interaction with His 94 and Trp 210. Additionally, it formed pi–alkyl interaction with Val 130 and Val 121 and pi–pi T-shaped interaction with His 94. It exhibited van der Waals interactions with Gln 92, Gln 71, Val 142, Glu 106, His 96, and Thr 201.

Compound **10d** binds to hCA IX through Zn (II) attractive charge interaction along with hydrogen bonds with Thr 200, His 96, and His 86 (Figure 8), as well as pi–alkyl interaction with Leu 199 and alkyl interaction with His 94. This showed van der Waals interaction with Val 121, Thr 200, Trp 210, Glu 106, and His 119.

Compound **10l** binds to hCA IX through hydrogen bonds with His 68, Gln 71, and Gln 92. Additionally, it exhibited pi–alkyl interaction with Val 130. It showed van der Waals interactions with crucial amino acids such as Leu 91, His 96, Ser 69, His 94, Thr 200, Leu 199, and Val 121.

Compound **10f** binds to hCA IX through Zn ion and forms hydrogen bonds with Thr 201, Trp 9, His 68, and His 96 (Figure 8). Moreover, it exhibited pi–alkyl interaction with His 94, Leu 199, Val 121, and Trp 210, and van der Waals interaction with Ser 69, Pro 202, Gln 71, Val 121, Asp 131, and Gln 92.

### 2.4. SAR Study

Uracil is well documented as a metal-binding pharmacophore [89,90,91] with particular emphasis on its Zn^+2^-binding abilities [92,93]. In the present work, uracil has been established as a bioisostere of the zinc-binding moiety benzenesulfonamide in our synthesized carbonic anhydrase inhibitors **10a–l**. The uracil Zn-binding ability has been proven using modeling studies and realized by the biological screening against CA isoforms (I, II, IX, XII) and cancer cell lines.

The SAR of the synthesized candidates can be summarized as follows (Figure 9):-Both uracil and thiouracil had CA inhibitory activity.-Substitution on uracil N-1 with a bulky group (benzyl **10b** and **10i**) decreases activity.-Substitution on the quinoline ring has tolerable activity, but greatly improves the selectivity, particularly when in combination with thiouracil (**10l**).

## 3. Materials and Methods

### 3.1. Chemistry

Melting points were determined with a Gallenkamp (London, UK) melting point apparatus and were uncorrected. ^1^H NMR and ^13^C NMR spectra were recorded on a Varian Gemini-400 (400MHz, Foster City, Calif., USA) spectrometer using chloroform (CDCl_3_)_,_ dimethylsulphoxide and/or (DMSO/D_2_O) as solvents and tetramethylsilane (*TMS*) as an internal standard (chemical shift in δ, ppm). ^1^H NMR data were recorded as follows: chemical shift (δ) [multiplicity, coupling constant(s) J (Hz), relative integral], where multiplicity is defined as s = singlet, d = doublet, t = triplet, q = quartet, and m = multiplet, or combinations of the above. High-resolution measurements were conducted on a time-of-flight instrument. All the results of elemental analyses corresponded to the calculated values within experimental error. The reaction progress was observed by thin-layer chromatography (TLC) using TLC sheets precoated with ultraviolet (UV) fluorescent silica gel (Merck 60F254) and spots were visualized by irradiation with UV light (254 nm) or iodine vapors. All starting materials and reagents were generally commercially available and purchased from Sigma-Aldrich or Lancaster Synthesis Corporation (Lancaster, UK). Compounds **4a–c** and **9a–e** were prepared according to the reported method [80,81,94,95,96].

#### 3.1.1. General Procedures for the Preparation of **10a–l**

A mixture of 5,6-diamino-uracils/thiouracils (1.28 mmol) and quinolone carbaldehydes (1.28 mmol) in ethanol (50 mL) was heated under reflux for 1 h. Cool the reaction mixture, the formed precipitate was filtered, washed with ethanol and crystallized from DMF/ethanol (1:1).

##### (*E*)-6-Amino-5-(((2-oxo-1,2-dihydroquinolin-3-yl)methylene)amino)pyrimidine-2,4(1*H*,3*H*)-dione (**10a**)

Yellowish white solid, Yield: 90%; m.p.: 270–272 °C; ^1^H NMR (DMSO-*d*_6_, 400 MHz) δ: 7.17–7.30 (m, 3H, 1Ar-H+NH_2_), 7.36 (d, *J* = 8.3 Hz, 1H), 7.44–7.47 (s, 1H, 1ArH), 7.64–7.70 (m, 1H, ArH), 7.91 (s, 1H, CH quinoline), 8.51 (s, 1H, CH=N), 8.74 (s, 1H, NH uracil), 9.75 (s, 1H, NH quinoline), 10.24 (s, 1H, NH uracil) ppm. ^13^C NMR (DMSO-*d*_6_, 100 MHz) δ: 189.78, 161.75, 159.39, 142.44, 138.66, 133.69, 130.92, 129.29, 128.42, 125.60, 122.07, 119.63, 115.41, 99.46 ppm (Appendix A). MS: m/z (rel. int.) = 297 (M^+^, 10), 145 (49.00), 111 (35.00), 44 (100.00) (Appendix A). Anal. Calcd for C_14_H_11_N_5_O_3_: C, 56.57; H, 3.73; N, 23.56; Found: C, 56.79; H, 3.89; N, 23.34%.

##### (*E*)-6-Amino-1-benzyl-5-(((2-oxo-1,2-dihydroquinolin-3-yl)methylene)amino) pyrimidine-2,4(1*H*,3*H*)-dione (**10b**)

Yellowish white solid, Yield: 80%; m.p.: 280–282 °C; ^1^H NMR (DMSO-*d*_6_, 400 MHz) δ: 5.24 (s, 2H, CH_2_-benzyl), 7.16–7.20 (m, 1H, Ar-H), 7.26–7.31 (m, 4H, Ar-H), 7.35–7.39 (m, 2H, Ar-H), 7.45–7.49 (m, 3H, 1ArH+NH_2_), 7.70 (d, *J* = 7.3 Hz, 1H, ArH), 8.80 (s, 1H, CH quinoline), 9.84 (s, 1H, CH=N), 10.93 (s, 1H, NH quinoline), 11.88 (s, 1H, NH uracil) ppm. ^13^C NMR (DMSO-*d*_6_, 100 MHz) δ: 161.78, 157.91, 154.40, 149.38, 144.34, 138.73, 136.20, 134.02, 129.14, 128.56, 127.27, 126.39, 122.08, 119.64, 114.95, 100.09, 44.57 ppm (Appendix A). MS: m/z (rel. int.) = 387 (M^+^, 6.00), 77 (83.00), 43 (99.00), 91 (100.00) (Appendix A). Anal. Calcd for C_21_H_17_N_5_O_3_: C, 56.11; H, 4.42; N, 18.08; Found: C, 65.29; H, 4.57; N, 18.26%.

##### (*E*)-6-Amino-1-ethyl-5-(((2-oxo-1,2-dihydroquinolin-3-yl)methylene)amino)pyrimidine-2,4(1*H*,3*H*)-dione (**10c**)

Yellowish solid, Yield: 87%; m.p.: 290–292 °C; ^1^H NMR (DMSO-*d*_6_, 400 MHz) δ: 1.17 (t, *J* = 6.9 Hz, 3H, CH_3_), 3.97 (q, *J* = 6.7 Hz, 2H, CH_2_), 7.18–7.48 (m, 3H, Ar-H), 7.49 (s, 2H, NH_2_), 7.73 (d, *J* = 7.7 Hz, 1H, ArH), 8.81 (s, 1H, CH quinoline), 9.81 (s, 1H, CH=N), 10.47 (s, 1H, NH quinoline), 11.87 (s, 1H, NH uracil) ppm. ^13^C NMR (DMSO-*d*_6_, 100 MHz) δ: 161.83, 157.87, 154.15, 148.98, 143.96, 138.70, 133.90, 130.31, 129.24, 128.64, 122.12, 119.70, 114.98, 99.99, 37.17, 13.08 ppm (Appendix A). MS: m/z (rel. int.) = 325 (M^+^, 18), 278 (40.00), 131 (70.00), 100 (84.00), 40 (100.00) (Appendix A). Anal. Calcd for C_16_H_15_N_5_O_3_: C, 59.07; H, 4.65; N, 21.53; Found: C, 59.31; H, 4.79; N, 21.69%.

##### (*E*)-6-Amino-1-methyl-5-(((2-oxo-1,2-dihydroquinolin-3-yl)methylene)amino) pyrimidine-2,4(1*H*,3*H*)-dione (**10d**)

Yellow solid, Yield: 90%; m.p.: 298–300 °C; ^1^H NMR (DMSO-*d*_6_, 400 MHz) δ: 3.35 (s, 3H, CH_3_), 7.18–7.22 (m, 1H, Ar-H), 7.30 (d, *J* = 8.2 Hz, 1H, ArH), 7.45–7.50 (s, 3H, 1ArH+NH_2_), 7.73 (d, *J* = 7.0 Hz, 1H, ArH), 8.82 (s, 1H, CH quinoline), 9.82 (s, 1H, CH=N), 10.75 (s, 1H, NH quinoline), 11.87 (s, 1H, NH uracil) ppm. ^13^C NMR (DMSO-*d*_6_, 100 MHz) δ: 161.76, 157.80, 155.07, 149.21, 143.76, 138.66, 133.74, 130.19, 129.24, 128.54, 122.02, 119.65, 114.91, 100.03, 29.45 ppm (Appendix A). MS: m/z (rel. int.) = 311 (M^+^, 30), 233 (100.00), 163 (61.00), 138 (43.00), 54 (50.00) (Appendix A). Anal. Calcd for C_15_H_13_N_5_O_3_: C, 57.87; H, 4.21; N, 22.50; Found: C, 58.04; H, 4.37; N, 22.32%.

##### (*E*)-3-(((6-Amino-1-methyl-4-Oxo-2-thioxo-1,2,3,4-tetrahydropyrimidin-5-yl)imino)methyl) quinolin-2(1*H*)-one (**10e**)

Yellowish solid, Yield: 85%; m.p.: 284–286 °C; ^1^H NMR (DMSO-*d*_6_, 400 MHz) δ: 3.85 (s, 3H, N-CH_3_), 7.19–7.32 (m, 2H, Ar-H), 7.48–7.52 (m, 1H, ArH), 7.60 (s, 2H, NH_2_), 7.74 (d, *J* = 7.1 Hz, 1H, ArH), 8.90 (s, 1H, CH quinoline), 9.90 (s, 1H, CH=N), 11.92 (s, 1H, NH quinoline), 12.22 (s, 1H, NH uracil) ppm. ^13^C NMR (DMSO-*d*_6_, 100 MHz) δ: 173.72, 161.67, 155.12, 154.31, 146.77, 138.95, 134.90, 130.61, 128.77, 128.75, 122.12, 119.53, 115.00, 103.79, 36.44 ppm (Appendix A). MS: m/z (rel. int.) = 327 (M^+^, 21), 297 (53.00), 242 (80.00), 228 (100.00), 197 (134.00) (Appendix A). Anal. Calcd for C_15_H_13_N_5_O_2_S: C, 55.04; H, 4.00; N, 21.39; Found: C, 55.27; H, 4.18; N, 21.53%.

##### (*E*)-6-Amino-1-ethyl-5-(((8-methyl-2-oxo-1,2-dihydroquinolin-3-yl)methylene) amino)pyrimidine-2,4(1*H*,3*H*)-dione (**10f**)

Yellow solid, Yield: 82%; m.p.: 278–280 °C; ^1^H NMR (DMSO-*d*_6_, 400 MHz) δ: 1.17 (t, *J* = 7.0 Hz, 3H, CH_3_), 2.44 (s, 3H, CH_3_), 3.97 (q, *J* = 6.7 Hz, 2H, CH_2_), 7.10–7.14 (m, 1H, Ar-H), 7.33 (d, *J* = 7.2 Hz, 1H, ArH), 7.50 (s, 2H, NH_2_), 7.60 (d, *J* = 7.7 Hz, 1H, ArH), 8.81 (s, 1H, CH quinoline), 9.84 (s, 1H, CH=N), 10.75 (s, 1H, NH quinoline), 11.01 (s, 1H, NH uracil) ppm. ^13^C NMR (DMSO-*d*_6_, 100 MHz) δ: 162.23, 157.84, 154.10, 148.92, 143.79, 137.06, 134.48, 131.55, 128.83, 126.78, 123.28, 121.85, 119.70, 99.92, 36.47, 17.19, 13.04 ppm (Appendix A). MS: *m*/*z* (rel. int.) = 339 (M^+^, 10), 294 (38.00), 239 (37.00), 105 (62.00), 77 (100.00), 44 (88.00) (Appendix A). Anal. Calcd for C_17_H_17_N_5_O_3_: C, 60.17; H, 5.05; N, 20.64; Found: C, 60.38; H, 5.27; N, 20.89%.

##### (*E*)-3-(((6-Amino-1-methyl-4-oxo-2-thioxo-1,2,3,4-tetrahydropyrimidin-5-yl) imino)methyl)-8-methylquinolin-2(1*H*)-one (**10g**)

Straw yellow solid, Yield: 78%; m.p.: 265–267 °C; ^1^H NMR (DMSO-*d*_6_, 400 MHz) δ: 2.44 (s, 3H, CH_3_), 3.85 (s, 3H, N-CH_3_), 7.11–7.15 (m, 1H, Ar-H), 7.35 (d, *J* = 7.2 Hz, 1H, ArH), 7.60–7.62 (m, 3H, 1ArH+NH_2_), 8.90 (s, 1H, CH quinoline), 9.91 (s, 1H, CH=N), 11.07 (s, 1H, NH quinoline), 12.22 (s, 1H, NH uracil) ppm. ^13^C NMR (DMSO-*d*_6_, 100 MHz) δ: 174.18, 162.62, 155.61, 154.80, 147.18, 137.81, 136.04, 132.42, 128.85, 127.44, 123.85, 122.42, 120.07, 104.25, 36.93, 17.65 ppm (Appendix A). MS: m/z (rel. int.) = 341 (M^+^, 14), 225 (24.00), 172 (100.00), 161 (83.00) (Appendix A). Anal. Calcd for C_16_H_15_N_5_O_2_S: C, 56.29; H, 4.43; N, 20.51; Found: C, 56.43; H, 4.60; N, 20.46%.

##### (*E*)-6-Amino-5-(((6-methoxy-2-oxo-1,2-dihydroquinolin-3-yl)methylene)amino) pyrimidine-2,4(1*H*,3*H*)-dione (**10h**)

Faint yellow solid, Yield: 85%; m.p.: 275–277 °C; ^1^H NMR (DMSO-*d*_6_, 400 MHz) δ: 3.80 (s, 3H, OCH_3_), 7.11–7.32 (m, 5H, 3Ar-H+NH_2_), 7.47 (s, 1H, CH quinoline), 8.45 (s, 1H, CH=N), 8.70 (s, 1H, NH uracil), 9.74 (s, 1H, NH quinoline), 10.24 (s, 1H, NH uracil) ppm. ^13^C NMR (DMSO-*d*_6_, 100 MHz) δ: 164.41, 161.35, 159.39, 154.20, 133.20, 132.87, 129.79, 120.25, 119.48, 116.18, 109.28, 99.70, 55.34 ppm (Appendix A). MS: m/z (rel. int.) = 327 (M^+^, 5), 160 (50.00), 104 (62.00), 43 (100.00) (Appendix A). Anal. Calcd for C_15_H_13_N_5_O_4_: C, 55.05; H, 4.00; N, 21.40; Found: C, 55.32; H, 4.21; N, 21.79%.

##### (*E*)-6-Amino-1-benzyl-5-(((6-methoxy-2-oxo-1,2-dihydroquinolin-3-yl)methylene) amino) pyrimidine-2,4(1*H*,3*H*)-dione (**10i**)

Yellowish solid, Yield: 88%; m.p.: 291–293 °C; ^1^H NMR (DMSO-*d*_6_, 400 MHz) δ: 3.80 (s, 3H, OCH_3_), 5.25 (s, 2H, CH_2_-benzyl), 7.13–7.21 (m, 2H, Ar-H), 7.24–7.31 (m, 4H, Ar-H), 7.36–7.40 (m, 2H, Ar-H), 7.46 (s, 2H, NH_2_), 8.76 (s, 1H, CH quinoline), 9.84 (s, 1H, CH=N), 10.94 (s, 1H, NH quinoline), 11.80 (s, 1H, NH uracil) ppm. ^13^C NMR (DMSO-*d*_6_, 100 MHz) δ: 161.32, 157.88, 154.39, 154.23, 149.35, 144.50, 136.21, 133.60, 133.38, 129.42, 128.54, 127.25, 126.39, 120.16, 119.73, 116.25, 109.51, 100.13, 55.40, 44.58 ppm (Appendix A). MS: m/z (rel. int.) = 417 (M^+^, 15), 361 (32.00), 193 (42.00), 161 (48.00), 109 (100.00) (Appendix A). Anal. Calcd for C_22_H_19_N_5_O_4_: C, 63.30; H, 4.59; N, 16.78; Found: C, 63.57; H, 4.70; N, 16.94%.

##### (*E*)-6-Amino-1-ethyl-5-(((6-methoxy-2-oxo-1,2-dihydroquinolin-3-yl)methylene)amino)pyrimidine-2,4(1*H*,3*H*)-dione (**10j**)

Yellowish solid, Yield: 79%; m.p.: 287–289 °C; ^1^H NMR (DMSO-*d*_6_, 400 MHz) δ: 1.17 (t, *J* = 6.9 Hz, 3H, CH_3_), 3.80 (s, 3H, OCH_3_), 3.98 (q, *J* = 6.7 Hz, 2H, CH_2_), 7.12–7.25 (m, 3H, Ar-H), 7.47 (s, 2H, NH_2_), 8.76 (s, 1H, CH quinoline), 9.82 (s, 1H, CH=N), 10.75 (s, 1H, NH quinoline), 11.78 (s, 1H, NH uracil) ppm. ^13^C NMR (DMSO-*d*_6_, 100 MHz) δ: 161.46, 157.92, 154.37, 154.21, 149.04, 144.23, 133.57, 133.41, 129.55, 120.30, 119.82, 116.37, 109.65, 100.10, 55.53, 37.25, 13.12 ppm (Appendix A). MS: m/z (rel. int.) = 355 (M^+^, x11), 298 (86.00), 282 (86.00), 246 (29.00), 196 (100.00) (Appendix A). Anal. Calcd for C_17_H_17_N_5_O_4_: C, 57.46; H, 4.82; N, 19.71; Found: C, 57.70; H, 4.95; N, 19.97%.

##### (*E*)-6-Amino-5-(((6-methoxy-2-oxo-1,2-dihydroquinolin-3-yl)methylene)amino)-1-methyl pyrimidine-2,4(1*H*,3*H*)-dione (**10k**)

Whitish solid, Yield: 89%; m.p.: 293–295 °C; ^1^H NMR (DMSO-*d*_6_, 400 MHz) δ: 3.35 (s, 3H, N-CH_3_), 3.81 (s, 3H, OCH_3_), 7.12–7.25 (m, 3H, Ar-H), 7.42 (s, 2H, NH_2_), 8.77 (s, 1H, CH quinoline), 9.82 (s, 1H, CH=N), 10.75 (s, 1H, NH quinoline), 11.78 (s, 1H, NH uracil) ppm. ^13^C NMR (DMSO-*d*_6_, 100 MHz) δ: 161.34, 159.42, 157.81, 155.08, 154.23, 149.22, 143.98, 133.33, 129.53, 120.19, 119.67, 116.24, 109.47, 100.10, 55.42, 29.49 ppm (Appendix A). MS: m/z (rel. int.) = 341 (M^+^, 53), 288 (55.00), 238 (82.00), 177 (48.00), 56 (100.00) (Appendix A). Anal. Calcd for C_16_H_15_N_5_O_4_: C, 56.30; H, 4.43; N, 20.52; Found: C, 56.48; H, 4.50; N, 20.81%.

##### (*E*)-3-(((6-Amino-1-methyl-4-oxo-2-thioxo-1,2,3,4-tetrahydropyrimidin-5-yl)imino)methyl)-6-methoxyquinolin-2(1*H*)-one (**10l**)

Yellowish solid, Yield: 84%; m.p.: 295–297 °C; ^1^H NMR (DMSO-*d*_6_, 400 MHz) δ: 3.81 (s, 3H, N-CH_3_), 3.85 (s, 3H, OCH_3_), 7.17–7.26 (m, 3H, Ar-H), 7.58 (s, 2H, NH_2_), 8.85 (s, 1H, CH quinoline), 9.89 (s, 1H, CH=N), 11.83 (s, 1H, NH quinoline), 12.22 (s, 1H, NH uracil) ppm. ^13^C NMR (DMSO-*d*_6_, 100 MHz) δ: 173.71, 161.24, 155.11, 154.31, 154.26, 146.98, 134.47, 133.64, 129.05, 120.15, 120.06, 116.34, 109.57, 103.86, 55.42, 36.48 ppm (Appendix A). MS: m/z (rel. int.) = 357 (M^+^, 20), 77 (94.00), 58 (100.0), 43 (85.00) (Appendix A). Anal. Calcd for C_16_H_15_N_5_O_3_S: C, 53.77; H, 4.23; N, 19.60; Found: C, 53.89; H, 4.41; N, 19.87%.

### 3.2. Biology

All adopted procedures for the conducted in vitro biological assays were performed as described earlier; CA (stopped-flow [4,30,36]), cytotoxicity (MTT [37,38]), and assessment of apoptotic markers [39,40] assays, as well as the induction of hypoxia with cobalt chloride [41,42]. Cancer cell line 250 (MCF-7) (HTB-22 from ATCC) and lung cancer cell line (A549) (CCL-185 from ATCC) used in this study were obtained from the VACSERA (Giza, Egypt) cell culture unit that was originally acquired from ATCC (Manassas, VA, USA) https://www.vacsera.com/ (accessed on 20 November 2021).

### 3.3. Computational Studies

#### 3.3.1. Molecular Modeling Study

The molecular modeling studies were fulfilled by the Molecular Operating Environment software (Discovery Studio). The crystal structure for hCA IX co-crystallized with 5-(1-naphthalen-1-yl-1,2,3-triazol-4-yl) thiophene-2-sulfonamide was downloaded from the Protein Data Bank (PDB ID: 5FL4) [87]. The protein was prepared for docking as follows: water molecules were ignored; hydrogen atoms were added; and the co-crystallized ligand was used to determine the binding pocket. The compounds were drawn on ChemDraw and transferred to Discovery Studio (Appendix A).

#### 3.3.2. Prediction of Pharmacokinetics Properties and Drug Likeliness

SwissADME server—a free web tool (http://www.swissadme.ch/index.php (accessed on 10 November 2021) developed by Swiss Institute of Bioinformatics—was utilized to compute physicochemical descriptors as well as to predict ADME parameters, pharmacokinetic properties, druglike nature, and the medicinal chemistry friendliness of compounds 6a–m to support drug discovery [86].

## 4. Conclusions

A novel set of quinoline-based uracil hybrids has been tailored and synthesized. The ability of the later hybrids **10a–l** toward inhibition of hCA I, II (cytosolic) and hCA IX, XII (transmembrane, tumor-associated isoforms) using colorimetric assay was evaluated. The results revealed that our novel hybrids **10a–l** had selective inhibition of hCA IX and XII comparable with hCA I and II in the micromolar range. Hybrids **10d** and **10l** have been carefully selected, as optimal compounds for higher hCA IX inhibitory activity and selectivity, for further investigation of their antiproliferative activity against the breast cancerous cell line MCF-7 and the lung cancer cell line A549 using the MTT protocol, which was comparable to the reference standard staurosporine. Compound **10d** had a superior inhibition of MCF-7 cells than A549 cells with IC_50_ = 2.87 ± 0.05 and 11.83 ± 0.22, respectively. Similarly, the hybrid **10l** displayed stronger inhibition of MCF-7 than of A549 with IC_50_ = 4.08 ± 0.08 and 26.10 ± 0.56, respectively. Further, both hybrids induced apoptosis in MCF-7 and A549 cells, together with worthy and desirable changes in Bax/Bcl expression ratio. The modeling study displayed high docking scores and good binding interactions of the most active compounds, **10d** and **10l**, within the hCA-IX active pocket, with adoption of orientation similar to that of co-crystalized ligand. This highlights our proposition of the impact of the electron-rich environment of the sulfur atom on the uracil backbone, as well as the insertion of small lipophilic and non-sterically hindered groups on both quinoline and uracil pharmacophores as crucial features for accessing highly selective hCA IX and XII inhibitors. Overall, our novel hybrids have opened the door to a new authentic approach for the engaging the quinoline scaffold with the uracil pharmacophore, which is a tactic that has rarely been discussed to date. Indeed, compounds **10d** and **10l** are likely to be potential lead candidates for further investigation and optimization, i.e., the rational development of novel, potent tumor-associated hCAs IX and XII selective inhibitors as agents for cancer treatment.

## Data Availability

The data is contained within the article and Appendix A.

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
