# Peer review of "Uracil as a Zn-Binding Bioisostere of the Allergic Benzenesulfonamide in the Design of Quinoline–Uracil Hybrids as Anticancer Carbonic Anhydrase Inhibitors"

_pharmaceuticals, 2022, doi:10.3390/ph15050494_

Round 1

Reviewer 1 Report

The manuscript presented by the authors deals with the evaluation of some hybrids of quinoline and pyrimidine as potential carbonic anhydrase inhibitors. In general, the authors' work is extensive and the classic and current methodologies within medicinal chemistry. However, before recommending the acceptance of the manuscript, the reviewer considers that it is necessary to clarify some ideas of the document:

1) it would be necessary to describe the justification for which an imine was selected as the linkage between both heterocyclic cores.

2) the inclusion of an specific discussion section would help the readers to have a better understanding of the finding of this study

3) Since only derivatives with S/O in position 2 of the pyrimidine core were evaluated, it is difficult to conclude that the presence of these atoms improves the activity

4) What evidence from spectroscopic data can confirm that imine formation ocurred at position 5 of the pyrimidine nucleus?

Author Response

Comments and Suggestions for Authors

The manuscript presented by the authors deals with the evaluation of some hybrids of quinoline and pyrimidine as potential carbonic anhydrase inhibitors. In general, the authors' work is extensive and the classic and current methodologies within medicinal chemistry. However, before recommending the acceptance of the manuscript, the reviewer considers that it is necessary to clarify some ideas of the document:

1) it would be necessary to describe the justification for which an imine was selected as the linkage between both heterocyclic cores.

Response: These hybrid structures have been lined chemically via an imine bond (Schiff’s base) based on the reported CA inhibitory activity of its derivatives (compound V in figure 4). A reference has been added (Arch Pharm Chem Life Sci. 2018;351:e1800146.)

2) the inclusion of an specific discussion section would help the readers to have a better understanding of the finding of this study

Response: We appreciate reviewer's valuable comment, we have followed the journal style of indicating results and discussion in one section as in our point of view will give a clear picture for each experiment to indicate the results and discuss it in the same part. The authors would like to keep the results and discussion as one section.

3) Since only derivatives with S/O in position 2 of the pyrimidine core were evaluated, it is difficult to conclude that the presence of these atoms improves the activity

Response: The comment about S/O impact on activity has been removed from figure 9.

4) What evidence from spectroscopic data can confirm that imine formation ocurred at position 5 of the pyrimidine nucleus?

Response: 

  1. The amine group at position 6 is less reactive and not expected to perform reactions with aldehyde as per the reported work (Int. J. Mol. Sci.202122(20), 10979; https://doi.org/10.3390/ijms222010979)
  2. The nucleophilic activity of the amino group of uracil moiety in position 6 is not active enough for the condensation reaction with quinolone-3-carbaldehydes in comparison with the amino group in 5 position 1,2 because it is present in enaminone form with carbonyl group of uracil in position 4 as shown below

Based on this fact, the condensation reaction of quinolone-3-carbaldehydes takes place with more easily nucleophilic aminouracils group in 5 position rather than the 6-amino group position. This fact explained by 1H-NMR which showed the appearance of a singlet signal at the range of δ 7.17 – 7.62 ppm for NH2 (6) protons and the disappearance of the protons of NH2 (5) at the range of δ 5.5 – 6.0 ppm. The NH(3) of uracils appears at δ 10.24-12.22 ppm depending on the N-R substituent and 2-oxo-or 2-thioxouracils.

References

  1. Hassan H. Hammud, Shawky El Shazly, Ghassan Sonji, Nada Sonji, Kamal H. Bouhadir, Thiophene aldehyde-diamino uracil Schiff base: A novel fluorescent probe for detection and quantification of cupric, silver and ferric ions, Spectrochimica Acta Part A: Molecular and Biomolecular Spectroscopy, Volume 150, 2015, Pages 94-103, https://doi.org/10.1016/j.saa.2015.05.038.

(https://www.sciencedirect.com/science/article/pii/S1386142515006265)

  1. Marx D, Wingen LM, Schnakenburg G, Müller CE and Scholz MS (2019) Fast, Efficient, and Versatile Synthesis of 6-amino-5-carboxamidouracils as Precursors for 8-Substituted Xanthines. Front. Chem. 7:56. doi: 10.3389/fchem.2019.00056

Reviewer 2 Report

The authors present a research article on uracil based compounds as a Zn-binding bioisostere of the allergenic benzenesulfonamide designing 2-Quinoline-Uracil hybrids as Carbonic Anhydrase Inhibitors acting on cancer.

The paper is of considerable interest, but there are several points that need to be addressed:

1) language quality: the article has numerous typos, grammatical imperfections, rather clumsy wording, and some sentences that are hard to read. I suggest a professional language editing service or a native speaker to address this issue.

2) Scheme 3 is of poor graphical quality (low resolution) and Table 4 is not completely readable. Figure 10. has some distorted words/structures.  Please provide high-quality figures/tables.

3) the introduction is focused on uracil/quinolines as core structures in drugs, this has rather little to do with the rest of the paper, I suggest shortening this section and focusing on Carbonic anhydrase inhibitors thus extending figure 5 and shrinking figures 1-4. 

4) which other zinc-binding groups are used for Carbonic anhydrase inhibitors? This point should be discussed also in light of HDAC inhibitors which are also containing Zinc. What about the selectivity of the compounds regarding other metal-containing enzymes/complexes?

5) hybrid compounds: I do not agree with how the authors use this term as they are not working on two distinct targets but various isoforms of the same target, thus I suggest avoiding this term. 

Author Response

The authors present a research article on uracil based compounds as a Zn-binding bioisostere of the allergenic benzenesulfonamide designing 2-Quinoline-Uracil hybrids as Carbonic Anhydrase Inhibitors acting on cancer.

The paper is of considerable interest, but there are several points that need to be addressed:

1) language quality: the article has numerous typos, grammatical imperfections, rather clumsy wording, and some sentences that are hard to read. I suggest a professional language editing service or a native speaker to address this issue.

Response: A native speaker has revised the manuscript linguistically.

2) Scheme 3 is of poor graphical quality (low resolution) and Table 4 is not completely readable. Figure 10. has some distorted words/structures.  Please provide high-quality figures/tables.

Response: Scheme 1 has been replaced by better graphical quality, Table 4 is now readable

And Figure 10. has been amended to higher quality

3) the introduction is focused on uracil/quinolines as core structures in drugs, this has rather little to do with the rest of the paper, I suggest shortening this section and focusing on Carbonic anhydrase inhibitors thus extending figure 5 and shrinking figures 1-4. 

Response: Figure 1-4 have been reduced to two figures only (figure 2 and 3) and so the related paragraphs have been reduced to two paragraphs only. In addition, a further paragraph and figure (figure 1) have been added that discussing about HDACIs and various chemical moieties that act as warhead via binding to HDAC’s Zn atom.

4) which other zinc-binding groups are used for Carbonic anhydrase inhibitors? This point should be discussed also in light of HDAC inhibitors which are also containing Zinc. What about the selectivity of the compounds regarding other metal-containing enzymes/complexes?

Response: Other Zn binding groups that act on CAs have been discussed in paragraph 3 at the introduction section. As well, further paragraph and figure (figure 1) have been added that discussing about HDACIs and various chemical moieties that act as warhead via binding to HDAC’s Zn atom.

5) hybrid compounds: I do not agree with how the authors use this term as they are not working on two distinct targets but various isoforms of the same target, thus I suggest avoiding this term. 

Response: The authors used the term “Molecular hybridization” as an indication to combine two pharmacophores in a new, single chemical entity. We do not consider various isoforms as different targets  i.e. we used hybrid molecules as chemical entities with two or more structural domains (Quinoline and Uracil) having different biological functions and dual activity, indicating that a hybrid molecule acts as two distinct pharmacophores.

Reviewer 3 Report

In this work, the authors synthetized a set of quinoline-uracil hybrids as carbonic anhydrase (CA) inhibitors. Moreover, the ADME profiles and the possible binding modes of the ligands at the CA active site were evaluated using computational approaches.  I have no observations regarding the chemical and biological parts. However, the authors will have to address the following issues in the computational section.

Major comment

In Figure 9A, it is clearly seen that the sulfonamide group of the reference compound has both hydrogens. Nevertheless, this should not be the case since this acid group must be ionized to interact with the catalytic Zn ion (doi: 10.1021/acs.jmedchem.5b01343). Although the pKa in aqueous solution of the sulfonamide is 8.62 (calculated with Chemicalize), PROPKA predicts a value between 2.69 and 4.18 for this compound at the binding site. Like the reference compound, the synthesized molecules should be in their ionized state because they also have an acidic group in their structure. Therefore, the authors should repeat the molecular docking of the compounds, considering the correct protonation states of the molecules.

Minor comments

(1) Line 131. Compound V instead of Compound IV.

(2) Some words are offset in Figures 5 and 10.

(3) Figure 6. I suggest using pIC50 instead of IC50 to highlight the importance of the most active compounds.

(4) The interaction of the molecules with the Zn ion is not depicted in Figure 9.

(5) Table 4 is incomplete.

Author Response

In Figure 9A, it is clearly seen that the sulfonamide group of the reference compound has both hydrogens. Nevertheless, this should not be the case since this acid group must be ionized to interact with the catalytic Zn ion (doi: 10.1021/acs.jmedchem.5b01343). Although the pKa in aqueous solution of the sulfonamide is 8.62 (calculated with Chemicalize), PROPKA predicts a value between 2.69 and 4.18 for this compound at the binding site. Like the reference compound, the synthesized molecules should be in their ionized state because they also have an acidic group in their structure. Therefore, the authors should repeat the molecular docking of the compounds, considering the correct protonation states of the molecules.

Response: the molecular docking of the compounds has been repeated where the correct protonation states of the molecules have been used. Please check Molecular docking study and figure 9.

 Minor comments

(1) Line 131. Compound V instead of Compound IV.

Response: Done as advised

(2) Some words are offset in Figures 5 and 10.

Response: Figure 5 and 10 have been amended.

(3) Figure 6. I suggest using pIC50 instead of IC50 to highlight the importance of the most active compounds.

Response: We prefer to keep using IC50 since when data has been converted to pIC50 the following figure has been generated

While the IC50 figure is as following

(4) The interaction of the molecules with the Zn ion is not depicted in Figure 9.

Response: The interaction of molecules with Zn is now demonstrated in figure 9.

(5) Table 4 is incomplete.

Response: Table 4 is now complete and readable.

Round 2

Reviewer 1 Report

The reviewer thanks the authors for attending all my concerns. Therefore, I recommend the acceptance of the manuscript in its current form.

Reviewer 2 Report

The author applied and answered most of my comments. Even though I do not fully agree with the use of the term "hybrid" I see a clear improvement in the manuscript, thus I can suggest acceptance after some further proofreading for typos.